# Antibodies against Anthrax Toxins: A Long Way from Benchlab to the Bedside

**DOI:** 10.3390/toxins14030172

**Published:** 2022-02-25

**Authors:** Arnaud Avril, Jean-Nicolas Tournier, Jean-Charles Paucod, Bénédicte Fournes, Philippe Thullier, Thibaut Pelat

**Affiliations:** 1Département Microbiologie et Maladies Infectieuses, Institut de Recherche Biomédicale des Armées, 91220 Brétigny-sur-Orge, France; jean-nicolas.tournier@intradef.gouv.fr (J.-N.T.); jcpaucod@yahoo.fr (J.-C.P.); pthullier@yahoo.com (P.T.); t.pelat@orange.fr (T.P.); 2Ecole du Val-de-Grâce, 75005 Paris, France; 3Laboratoire Français du Fractionnement et des Biotechnologies, 91940 Les Ulis, France; fournesb@lfb.fr

**Keywords:** antibody, anthrax, biodefense, clinical development, in vivo protection

## Abstract

Anthrax is an acute disease caused by the bacterium *Bacillus anthracis*, and is a potential biowarfare/bioterrorist agent. Its pulmonary form, caused by inhalation of the spores, is highly lethal and is mainly related to injury caused by the toxins secretion. Antibodies neutralizing the toxins of *B. anthracis* are regarded as promising therapeutic drugs, and two are already approved by the Federal Drug Administration. We developed a recombinant human-like humanized antibody, 35PA83 6.20, that binds the protective antigen and that neutralized anthrax toxins in-vivo in White New Zealand rabbits infected with the lethal 9602 strain by intranasal route. Considering these promising results, the preclinical and clinical phase one development was funded and a program was started. Unfortunately, after 5 years, the preclinical development was cancelled due to industrial and scientific issues. This shutdown underlined the difficulty particularly, but not only, for an academic laboratory to proceed to clinical development, despite the drug candidate being promising. Here, we review our strategy and some preliminary results, and we discuss the issues that led to the no-go decision of the pre-clinical development of 35PA83 6.20 mAb. Our review provides general information to the laboratories planning a (pre-)clinical development.

## 1. Introduction

Anthrax is a lethal disease caused by the spore-forming, Gram-positive bacterium *Bacillus anthracis* that can infect humans [1]. Four forms of anthrax have been described so far: cutaneous, intestinal, inhalation, and injection anthrax. Each of these are lethal for humans in absence of treatment. Anthrax toxins and the antiphagocytic polyglutamic capsule are the two major virulence factors of *B. anthracis* ([2]. These virulence factors are encoded by genes located in the plasmids PXO1 and PXO2, respectively. The lethal toxin (LT) is composed of the Protective Antigen (PA) and the Lethal Factor (LF). The Edema Toxin (ET) is composed of the PA and the Edema Factor (EF). Both toxins penetrate into the cell by endocytosis. PA is involved in the binding of the toxin at the cell surface. LF and EF escape the vesicle of endocytosis and reach their target in the cytosol. LF is a protease that catalyzes the hydrolysis of the MAPKK, resulting in cell death by apoptosis. EF is a calmodulin-dependent adenylate cyclase that greatly increases the level of cAMP in the cell. The increase of cAMP results in the deregulation of water homeostasis (inducing an edema) and imbalance the intracellular signalling pathways and impairs de macrophage function, allowing the bacteria to further evade the immune system.

When untreated, the human mean lethal dose (LD_50_) is estimated between 8000 and 10,000 inhaled spores [3]. Cutaneous anthrax represents 95% of natural contaminations but is the less lethal form of anthrax (1% lethality when efficiently treated by antibiotics). Pulmonary anthrax has a lethality over 80% (up to 100% if untreated), even if after the 2001 attacks the lethality was estimated to be about 45% of treated people [4]. The gastrointestinal anthrax develops after contaminated meat consumption. Finally, a fourth form of anthrax was more recently identified [5]. This form results from the injection of drug contaminated by spores. According to its lethality and potential utilization as a biowarfare/bioterrorism agent, anthrax is classified among the six biological agents of the Center for Disease Control and Prevention (CDC) A-list. In the past several countries such as the Union of Soviet Socialist Republics, Great Britain, Canada, the United States of America, and Iraq were proven or suspected by the international community to weaponized anthrax. *B. anthracis* can be used as a biological weapon, as seen during the 2001 United States (US) postal service dissemination, when limitations in the current prophylaxis and curative protocols were unfortunately experienced [6].

Currently, anthrax therapy is mainly based on a 60-day administration of ciprofloxacin or doxycycline. While antibiotics can overcome bacteremia caused by antibiotic-susceptible strains of anthrax, they do not directly address the toxemia that drives pathogenesis. Additional limitations of antibiotics include poor patient compliance with the 60-day schedule and inefficacy against strains of *B. anthracis* that could became intentionally or naturally resistant to antibiotics. During the 2001 US anthrax dissemination, when at least 11 people were contaminated, five died despite treatment including antibiotics and intensive care. This outcome is consistent with the findings of previous studies, showing that the course of anthrax may progress to a point where the levels of secreted anthrax toxins are such that death is inevitable, even in the presence of efficient antibiotics. As the anthrax toxins are essential for lethality, they are targets of particular interest for therapeutic antibodies [7]. Indeed, antibodies can bind toxins in the bloodstream with a high affinity (nano- or pico-molar affinity), which preclude dissociation of the antibody-toxin complex until elimination by the immune system (phagocytosis…). Depending on the epitope targeted, the antibody can neutralize the toxin’s activity directly or indirectly by several mechanisms. Three anthrax antitoxins have been approved by the U.S. Food and Drug Administration (FDA): two of them are monoclonal antibodies (raxibacumab, and obiltoxaximab –anthim-), and the third one is human polyclonal purified IgG from vaccinated humans (intravenous anthrax immune globulin AIG-IV, also referred as anthrasil). These antibodies are stockpiled in the USA, in case of a bioterrorist attack. The CDC recommends the administration of both antibiotics and antibodies, despite recent studies questioning the efficiency of antibodies [8,9,10,11]. A subunit vaccine, BioThrax, is also approved by FDA for persons at high risk of exposure and for people potentially exposed to anthrax. BioThrax is composed chiefly of Protective Antigen (PA) and in a less extend of EF and LF. The US federal government has a goal to stockpile 75 million doses of BioThrax. The European Medicines Agency (EMA) approved raxibacumab as orphan and pediatric medicine for post-exposure prophylaxis of inhalational anthrax. EMA also approved Obiltoxaximab in 2020 as orphan medicine in the same indication.

In our laboratory, germlined-humanized anti-PA 83 kDa and anti-lethal factor (LF) antibodies were previously developed from macaques. We review here the development of these antibodies in a clinical perspective. First, the anti-PA “35PA83” antibody was isolated. The pre-clinical and clinical phase one development of this antibody was funded and started, but was cancelled after 5 years of development due to several industrial and scientific issues. This review summarizes the successes and the difficulties encountered during the research and clinical development. Furthermore, we present some in-vivo original data. The IgG 35PA83 protective efficacy was assessed in the White New Zealand (WNZ) rabbit model infected by a lethal strain of *B. anthracis* (the 9602 strain), in a biosafety level three laboratory. The IgG was tested in a prophylaxis schema, injected before the challenge by 100 LD_50_ of the 9602 strain, and as therapeutic, injected alone 6 h after the challenge (80 LD_50_ and 200 LD_50_). Then, the anti-LF “2LF” protective antibody was isolated using similar strategy, and was also engineered by hyper-germline humanization process [12].

## 2. Results

### 2.1. Pipeline Used for the Research and Development of Human-like Recombinant Antibodies

A pipeline for the isolation of recombinant antibodies was developed in our lab (Figure 1) [13]. Because antibodies will be administrated in human, they have to be well-tolerated. We previously demonstrated that antibodies of macaque origin are closed to human antibodies: the overall identity of the macaque variable heavy (VH) and variable light (VL) domain sequences with their most similar, human, germline gene counterpart is 78.5%, so they are predicted to be well-tolerated [14]. Because macaques are a model of choice for the development of human-like therapeutic antibodies, they were immunized with recombinant surface antigen or with non-toxic toxin (inactivated toxin or toxin subunit). Generally, three antigen administrations were realized at a 1-month interval [13]. After a resting period of at least 120 days after the last immunization, bone marrow was sampled, total RNA extracted and a RT-PCR was done to amplify the DNA region coding the antibody variable domains. After this resting period, no significant amplification of the gene coding for the antibody variable genes is expected in RT-PCR, meaning that the macaque’s immune response has returned to background level. A final boost was then realized after this control and bone marrow was iteratively sampled with a 2-to-3-day frequency until a decrease in the immune response was observed. Polymerase Chain Reaction (PCR) products obtained when the immune response was optimal were pooled and used for the generation of a phage-display single chain Fragment variable (scFv) or Fragment antigen binding (Fab) library in pHAL14, pHAL32, or pComb3X phagemid vector [12,13,15,16]. The resulting libraries were screened against the recombinant protein or the whole pathogen of interest. After three to four rounds of phage-display screening, (sub-)nanomolar scFv/Fab are expected. The DNAs of the isolated scFv/Fabwere sequenced and non-redundant sequences were expressed as soluble scFv/Fab then purified by chromatography. All purified antibody fragments were then tested in ELISA and in surface plasmon resonance (SPR, Biacore^®^ technology, Uppsala, Sweden) to determine their reactivity/affinity. scFv/Fab with affinity better than 10 nM for the target of interest were characterized in relevant in-vitro or ex-vivo neutralising assay. The best candidates, regarding neutralization, were finally expressed as full-length IgG and tested in relevant in-vivo protection assay. The most promising IgG is then selected as the best hit for potential clinical development. The other unselected IgG, having good affinity (i.e., nanomolar or better), could be used for the development of detection assay/diagnostic kits. Even if we demonstrated the proximity between human and macaque antibodies, differences existing between human and macaque variable (V) genes are four-fold greater than the inter-human polymorphism, and such differences may be immunogenic. Indeed, in our lab we used a germline-humanization approach, based on results raised using two mathematical predictive indices: the H- and the G-score [17]. H- and G-scores show that V regions from macaque light chains may sometimes be indistinguishable from their human counterparts, but this was not the case for V regions from macaque heavy chains indicating that humanization step is required. The “germline humanization” utilizes human germline sequences as templates for humanization instead of the more frequently utilized sequences of human-expressed IgGs. It was shown that germline humanization significantly improved both the G- and H-score. We previously successfully used the super-humanization (mutation of amino acids in the Framework regions –FR-) and the hyper-humanization (mutation of amino acids in both FR and in the complementary determining regions –CDR-) for the humanization of the antibodies that we developed.

This pipeline was successfully used for the development of antibodies against different biological agent such as botulinum neurotoxins [18], ricin [19,20,21], haemorrhagic fevers [22,23] or anthrax toxins [12,24] and is currently used for the development of anti-orthopoxviruses antibodies.

### 2.2. Development of Anti-Anthrax Antibodies

#### 2.2.1. Development of an Anti-PA Antibody

The pipeline previously described was used for the development of an anti-PA antibody [24]. Indeed, PA is essential for the formation of both Lethal Toxin (LT) and Edema Toxin (ET) toxins. Targeting this subunit would thus neutralize simultaneously the activity of both toxins and would be most efficient in clinic. A macaque was immunized with four injections of 200 µg of recombinant PA83, in complete Freund’s adjuvant, until a titer of 1/100,000 was obtained. After the last boost, the modulation of the immune-response was observed. The total RNA was extracted from the bone marrow and retro-amplified with primers specific of the DNA encoding the κ and γ1-chains. The PCR products obtained with the sample where the immune response was maximal were cloned into pComb3X phagemid. This resulting Fab library was then panned by the phage-display technology against PA83 antigen. Five rounds of panning were realized, with increasing stringency. The DNA encoding the Fab isolated after the fifth round of panning was sequenced. Fifty of the unique sequences were expressed as soluble Fab and characterized in SPR and in an in-vitro neutralization assay based on the mouse macrophage cell line J774A.1 mortality assay.

With this strategy, the antibody 35PA83 was isolated. As Fab, 35PA83 had an IC_50_ of 5.6 +/− 0.13 nM in the cell-based neutralization-assay and in SPR an affinity for PA83 of 3.4 nM was measured with an off rate of 3.23 × 10^−4^ × s^−1^. As an IgG, 35PA83 had an affinity of 180 pM. The improvement of its affinity may be due to a higher stability of IgG compared to Fab.

#### 2.2.2. Development of an Anti-LF Antibody

All anti-anthrax toxin antibodies that are currently on the market target PA83. As anthrax is a biowarfare agent and was militarized in the past, it can be envisioned that PA would be intentionally mutated to make inefficient the antibodies currently developed. Even if this option is unlikely, the principle of biodefense is to be prepared for any scenario. As no anti-LF antibodies are currently marketed, the development of such antibodies could be of particular interest for biodefense. LF is the second crucial subunit of LT and targeting LF should be as efficient as targeting PA for LT neutralization. In-vivo data indicates that the neutralization of LT only would be sufficient to decrease the mortality, because LT is the main factor leading to death. In this context, anti-LF antibodies were similarly isolated in our lab [12].

A macaque was immunized three times to reach a titer of 1/400,000, resulting in the construction of an scFv phage-display library which was then screened against recombinant LF. After three rounds of panning, the eluted binders were sequenced and 40 non-redundant sequences were identified; among them was the best candidate, scFv 2LF, as scFv 2LF has an affinity of 1.02 nM. Interestingly, we demonstrated that the antibody 2LF also binds EF with an affinity of 5 nM, and as a consequence that the epitope of 2LF on LF is (at least partially) shared with EF [25]. In-vitro neutralization assay, 2LF has an half maximal inhibitory concentration (IC_50_) of 4 nM and in-vivo it was demonstrated that 10 µg of 2LF IgG completely inhibited the formation of edema induced by 20 µg of ET [25].

Each generation of therapeutic antibodies (murine, chimerized, humanized, and “fully-human”) were considered to be better tolerated than the previous one. We previously demonstrated that antibodies isolated from macaques were very close to human ones. Nevertheless, to be as safe as possible after administration to patient, particularly if several injections are required, we perform the germline-humanization of 2LF. This humanization step was realized firstly in the FR regions (“super-humanization”). The sequences of the human germline genes closest to the sequence of 2LF were identified with IMGT database. The amino acids diverging between the germline sequence and the VH and VL sequence of 2LF were identified and for each potential “mutation” an antibody variant was synthesized, produced and tested in surface plasmon resonance. As summarized in Table 1, only one mutation induces a slight decrease in affinity and was not retained. A mutant that combined the 28 selected mutations was produced for further characterization. The affinity of this mutant, called 2LFG1, was 0.7 nM, which is slightly better than the affinity of the parental antibody (1.02 nM), validating these 28 mutations. During the humanization, the germinality index (GI) was improved for both VH (81.3% to 98.9%) and VL (86.51% to 100%).

Because the highest tolerance is requested in clinic, we then went deeper in the humanization process. The humanization in the CDR, referred to as “hyper-humanization”, was also realized (Table 2). Five out of the nine amino acids diverging in the Complementary Determining Region (CDR) of 2LFG1 VH domain and three out of the six amino acids diverging in the CDR of 2LFG1 VL were efficiently humanized. A humanized antibody containing these eight additional amino acids in the CDR was expressed. The affinity of this antibody, called 2LFH1, was determined at 8.92 nM (8.7-fold higher than the parental scFv 2LF). 2LFH1 was selected for further development.

### 2.3. Filling the Gap between the Bench and the (Pre-)Clinical Development

#### 2.3.1. Intellectual Property Issue and Affinity Maturation

After its identification, 35PA83 was humanized and this work was published [26]. Unfortunately, the complete sequence of 35PA83 and of its humanized variant (hu4 35PA83) were included in the publication [24]. When the clinical development of this antibody was envisioned, it was not possible anymore to patent 35PA83 or hu435PA83, as the published sequences were then considered public. Such limitation would have been risky for an industrial development, because another company could have produced a copy-cat of the antibody, which was not strategically conceivable. To bypass this issue, we engineered the sequence of 35PA83 by performing an in vitro affinity maturation. For this reason, an antibody library composed of 4 × 10^8^ clones was generated by introducing controlled mutations on 34PA83 CDRs by using Massive Mutagenesis^®^ [27] approach and screened via optimized protocol against adsorbed, then soluble PA83 to specifically select high affinity binders. Using this strategy, we isolated the variant 35PA83 6.20, having an affinity of 180 pM, 19-fold improved by comparison to the parental 35PA83, mainly due to a decrease in its dissociation rate (koff = 0.51 × 10^−4^ s^−1^) [28]. The 35PA83 6.20 antibody sequence was patented and this variant was selected for (pre-)clinical development.

#### 2.3.2. Humanization of 35PA80 6.20

Because the affinity maturation consisted the introduction of mutations, the new antibody could have been potentially more immunogenic for humans. To improve the potential tolerance of 35PA83 6.20 antibody, we performed its super-humanization (Table 3, Figure 2). The closest human germline-gene sequences of 35PA83 6.20 sequences were identified with IMGT^®^ database and the amino acids diverging between the germline sequences and the 35PA3 6.20 sequences were defined. For each humanized residue (17 positions) an antibody variant was synthesized/produced and tested in ELISA and surface plasmon resonance. As many variants had to be screened, a limited number of dilutions were realized for each variant. As a consequence, relevant affinity (KD) was not calculated; the selection of variant was done on the basis of the Koff only. Only the three amino acids introduced during the affinity maturation processes were not mutated, because they were considered as crucial to preserve the good affinity. Finally, it was observed that seven mutations induced a decrease in affinity (Koff) or stability and were not retained. A mutant that combined the 10 selected mutations was produced. The affinity of this mutant, called 35PA83 6.20 G1, was 316 pM, which is quite similar to the affinity of the parental antibody (180 pM), confirming the 10 mutations. During the humanization, the germinality index was improved for both VH (89% to 91.5%) and VL (94.5% to 95.5%). Despite that we previously realized humanization of the CDR (“hyper-humanization”) of other antibodies, here, we considered that the benefit/risk balance was not in favor of the hyper-humanization. Indeed, a high GI was obtained already, and introduction of mutations in the CDR could decrease the antibody efficiency. In addition, as anthrax is not a chronic disease, only one or few antibody injections would be required; the risk of anti-drug antibody formation is thus unlikely. Furthermore, the hyper-humanization would have delayed the preclinical development of 35PA83 6.20.

#### 2.3.3. In-Vivo Protection Conferred by 35PA83

Because we initially planned the clinical development of 35PA83 6.20 and because this antibody could have been a national countermeasure, the in-vivo protection data had not been initially published.

The protection mediated by 35PA83 IgG was assessed in New Zealand rabbits infected with the virulent 9602 strain and tested in prophylaxis and therapeutic schemes. The protection mediated by 35PA83 IgG as a prophylaxis or as a therapy, alone or in combination with relevant antibiotherapy, was also assessed in Fisher 344 rat model intoxicated with LT and in A/J mice model infected by the vaccinal Sterne 7702 strain. To respect ethical animal experimentation, only a limited number of animals were used. Indeed, additional and more complete in-vivo experimentations were planned during the pre-clinical development (data not shown).

##### Pharmacokinetic Studies

The half-life of IgG 35PA_83_ was established as 94.6 h in WNZ rabbits (Table 4), in concordance with literature data. In WNZ rabbits (IgG injected subcutaneously, S.C.), the C_max_ is 36.9 µg·mL^−1^ and the AUC_0-∞_ is 3609 h·µg·mL^−1^. The values of the Mean Residence Time (MRT), Volume of distribution (VD) and Clearance (Cl) indicated that 35PA_83_ was slowly eliminated.

##### Prophylaxis and Delayed Treatment with IgG 35PA83 Administration Alone in Rabbit Model

The protective efficacy of the IgG 35PA_83_ was tested in the WNZ rabbit challenged with the virulent 9602 strain of B. anthracis. In prophylaxis, all the animals receiving a dose of 2.5 mg·kg^−1^ of IgG 35PA_83_ before an intranasal challenge with 100LD_50_ of spore survived (Figure 3) (significant effect versus positive control, *p* < 0.0001). Two of the eight animals receiving the IgG at a dose of 1 mg·mL^−1^ died at the fourth and the fifth day, when all untreated controls died after 48 h (significant effect versus positive control, *p*-value < 0.0001). Only one rabbit treated with the IgG at a dose of 0.5 mg·mL^−1^ survived, when the other died between the third and the sixth day (significant effect versus positive control, *p*-value < 0.0001).

The IgG 35PA_83_ was then tested for its capacity to protect the rabbits in a therapeutic scheme (Figure 4). When it was injected at the dose of 2.5 mg·mL^−1^ 6 h after the challenge with 80 LD_50_ of the 9602 spores, all the rabbits survived (significant effect versus untreated controls, *p* value < 0.0009). When the challenge was realized using 200 LD_50_, only 1 treated rabbit died after the fifth day (significant effect versus untreated controls, *p* value < 0.0004) when all the untreated control animals died between the 24th and the 48th hour.

Interestingly, anti-PA and anti-LF rabbit IgG were detected at a titer of 1280 to 2560 and of 320 to 640, respectively, in the sera sampled just before the animal euthanasia at the 21th day after the challenge, indicating that the animals started to develop their own natural immune responses.

### 2.4. Clinical Development of 35PA83 6.20

#### 2.4.1. Formulation and Stability

In 2011, a 10-year pre-clinical and clinical phase one development of 35PA83 6.20 was funded by the Direction Générale de l’Armement (under reference 2010.94.092). The development was sub-contracted with the Laboratoire Français du Fractionnement et des Biotechnologies (LFB). During the first step of clinical development, the antibody was expressed as IgG under Good Manufacturing Practices (GMP) conditions. The antibody was expressed in the rat YB2/0-E cell line, a property cell line of LFB. Several buffers were tested to ensure protein stability, and the buffer containing 30 mM acetate, 245 mM Mannitol and 400 ppm Polysorbate 80, pH 5.0 (+/−0.2) was selected. A concentration of 9.8 g·L^−1^ of 35PA83 6.20 was reached in this liquid buffer. A stability assay was realized in the final packaging by different methods: optical density (at 280 and 400 nM), dynamic light scattering, high pressure size exclusion chromatography, SDS-PAGE, reverse phase–high performance liquid chromatography, *Isoelectric focusing,* SPR and research of subvisible particles (data not shown). For each time-point, the sample was compared to the parental antibody (t = 0). In normal condition, no stability issue was observed after at least 24 months at 5 °C. This antibody would represent a medical countermeasure for the French Armed Forces, and could be shipped all over the world. During the transport, particularly in intertropical area, a cold chain failure could occur. A stability assay at +25 °C and +40 °C was thus also realized. Under these conditions, a stability issue was observed within 3 months. Nevertheless, when the drug product was exposed to +40 °C for 12 h to 48 h, no stability issue was observed, which is highly interesting for the military logistical chain.

#### 2.4.2. Scientific and Industrial Issue

During the preclinical development, a quality control was requested to approve a batch release. A complete characterization according to good manufacturing practices (affinity, epitope, pharmacokinetic…) of the antibody was performed, to be sure that it was completely functional. As the epitope, considered as linear, was initially identified with the pepscan technology [29], the confirmation was realized with the same approach. Because the funds available for the clinical development were significantly higher than those for the research, the epitope analysis was realized on the whole PA83 sequence instead only on the domain IV (the domain interacting with the receptor which is the main neutralizing site), and with a higher peptide resolution. During this control, the binding to previously identified peptide (“PLYISNPNY”, domain IV of PA83) was confirmed, but higher reactivity was observed for the YTVDVKNKRTFLSPWI region, which is localized in the domain I of PA83. The intensity of this binding was considered by pepscan as relevant for an epitope. This “new epitope” is close to the PA83 activation site. Indeed, once PA83 has bound the cell it recruits an heptamer of PA83. After heptamerization, PA83 is cleaved in PA23 + PA60 by furin. PA83 activation is essential for the penetration of LF and EF into the cell. The new localization of the epitope questions the mechanism of toxin-neutralization. Indeed, if the epitope is localized in domain one, it could be possible that 35PA83 6.20 binds circulating PA83 or PA83 after it have been connected to the cell. Unfortunately, 35PA83 6.20 is an IgG1 that has a low fucose rate (~35%), giving it high affinity for FcγRIIIAa (CD16a). A high affinity for this receptor is beneficial when the toxin is neutralized in the general circulation, because it allows a faster elimination of the toxin by the immune cells. If the antibody binds the PA83 after binding to the cell surface, it could also recruit cytotoxic cells that will lyse the cell. The consequence would be the death of the target cell due to an inappropriate immune activation instead of the anthrax toxin elimination (Figure 5). Additional antibody characterization experiments were realized to identify precisely the epitope, but they were not successful. A precise epitope mapping by mass spectrometry was planned, but another industrial issue occurred.

Most antibodies are expressed in CHO cell line, but in this project the LFB YB2/0-E cell line was used. During the antibody production process, Host Cell Proteins (HCP) were identified along with the antibody using standard recommended techniques and available software. Different options could have been studied further to remove the identified HCPs within the context of this particular project, but those were not pursued at the time due to specific constraints associated with the project.

The presence of HCPs represented a major issue for this industrial process. Different options have been studied but no appropriate solutions could be proposed. All options would have delayed the project for several years and would have requested additional funds.

#### 2.4.3. Commercial Issues

Three anthrax antitoxins have been approved by the U.S. Food and Drug Administration (FDA): two of them are monoclonal antibodies (raxibacumab and obiltoxaximab –anthim-), and the third is human polyclonal purified IgG from vaccinated humans (intravenous anthrax immune globulin AIG-IV, also referred as anthrasil).

During the preclinical development of 35PA83 6.20, the European Medicines Agency (EMA) approved raxibacumab as orphan and pediatric medicine for post-exposure prophylaxis of inhalational anthrax. Market exclusivity is an orphan incentive awarded by the European Commission to a specific clinical indication with an orphan designation. Each indication with an orphan designation confers ten years’ market exclusivity for the particular indication. As a consequence, to be approved by EMA, it would have been necessary to prove that 35PA83 6.20 confers significant advantage compared to raxibacumab, and approval would not be possible during the 10-year exclusivity.

Considering the scientific, industrial and commercial issues, the clinical development of 35PA83 6.20 was definitively stopped, to focus time and money on the development of other antibodies.

## 3. Discussion

In our laboratory, we developed a platform for the development of recombinant human-like antibodies. This platform includes the immunization of macaques with recombinant antigens, the phage-display screening, the in-vitro and in-vivo characterization of the antibodies, and the germline humanization of the best ones. This platform was successfully used for the development of antibodies directed against several biowarfare agent, such as anthrax. During the 2001 US anthrax dissemination, it was observed that people at risk of being exposed to anthrax would not take the full 60 days course of antibioprophylaxis, intended to prevent spore late germination and that, as foreseen but still very unfortunately, delayed treatments would not be able to successfully cure all patients. Therapeutic antibodies represent a drug of choice for the safe and specific treatment of anthrax, because only one or few administrations would be required. In this way we developed anti-anthrax antibodies targeting PA (35PA83 6.20) and LF (2LF). 35PA83 6.20 was a promising drug candidate and it was decided to undergo a clinical development. In the present study, the in-vivo efficacy of 35PA83 6.20 was assessed as a prophylaxis or a therapy, used at efficient and clinically relevant doses. Several animal models of anthrax infection have been used in previous studies, making it difficult to compare the results. The clinical presentation of inhalational anthrax in rabbits infected was comparable to that of inhalational anthrax in humans or in non-human primates, and thus these studies could provide the basis for approval anti-anthrax agents for use in humans when the use of non-human primate models is not possible. We then utilized the WNZ rabbit infected intranasally with the lethal 9602 strain, which was used to look up the protective efficiency of IgG 35PA83 6.20, particularly in a therapeutic scheme.

In this study, IgG 35PA83 was presented with standard criteria, allowing it to be compared with antibodies previously described in the literature. In a previous study [30], the affinity of one of the best antibody against PA83 (Ig 83K7C) was 118 pM, slightly better than IgG 35PA83 6.20 (180 pM). In the rat passive protection study, 0.3 nM IgG 83K7C was required for full protection versus 0.2 nM IgG 35PA83 6.20, in the same experimental conditions. IgG 35PA83 also compares favourably with various antibodies described in a second study, for which affinities of 82 to 711 pM were reported [31]. In this second study however, in-vitro and in-vivo (toxin-exposed rats) assays were realized under non-standard conditions, rendering it difficult for any further comparison with IgG 35PA83 6.20. In another study [32] fully human mAb anti-PA antibodies were described (MDX-1303). In the in-vitro neutralization assay, the IC_50_ of this antibody was 1 nM, whereas an IC_50_ of 0.75 nM for IgG 35PA83 6.20 was obtained in the same experimental conditions. One chimpanzee IgG, W1, with an affinity of 39.7 pM, was reported [33], conferred protection at low concentrations in the rat passive protection study, as all animals survived when this IgG was injected at a molar ratio of 1:4 (Ab:Protective Antigen). But these assays were realized under non usual conditions, with 7.5 µg of Protective Antigen (PA) when we used 40 µg, rendering it difficult for any further comparison with IgG 35PA83 6.20. We also compared 35PA83 6.20 to the two approved anti-anthrax antibodies; Raxibacumab (ABthrax) and Obilthoxaximab (Anthim). Raxibacumab is a fully human anti-PA antibody that has a neutralization mechanism similar to 35PA83 6.20. Its affinity for PA is 2.78 nM and its IC_50_ is 0.21 nM in the macrophage cell-killing assay. Raxibacumab neutralized the toxin by inhibiting the interaction between PA83 and its receptor. In the pivotal WNZ rabbit efficacy studies, none of the animals in the control groups (rabbit or nonhuman primate) challenged with 200 half lethal dose (LD_50_) aerosolized anthrax spores survived [34]. When rabbits were challenged with 200 LD_50_ of aerosolized anthrax spores and treated with Raxibacumab, 44.4% of animals treated with 40 mg·kg^−1^ survived. Likewise, in the pivotal study with nonhuman primates, 64.3% of those treated with 40 mg·kg^−1^ survived. Obiltoxaximab is an anti-PA IgG1 that was derived from murine monoclonal antibody 14B7 through modifications that included affinity enhancement, humanization, and deimmunization. Its affinity is 330 pM and has an IC50 of 0.08 mg·mL^−1^. WNZ rabbits were challenged with spore infection by inhalation of ~200 LD_50_ equivalents, before treatment with Obiltoxaximab. I.V. administration of 1, 4, 8 or 16 mg·kg^−1^ of antibody protects 17%, 33%, 69% and 62%, of the animals, respectively. Three studies were realized in a cynomolgus model, challenged in a similar fashion as the rabbit. Variable outcome was observed [35]. In a first study, survival rate for 4 or 8 mg·kg^−1^ were 79% and 73%, respectively, vs. 14% in the control group. In a second study, significant protection was only observed for an antibody dose of 16mg·kg^−1^ and when time to death is considered in the same time. In the last study there is no correlation between the antibody dose and the survival rate. Thus, IgG 35PA83 6.20 has affinity and neutralization properties at least equivalent to those of the best previously described antibodies.

The in-vivo protection of IgG 35PA83 6.20 was evaluated in three animal models of anthrax, but here, we only presented the data obtained with the WNZ rabbit model, which is more relevant than the rat and the mice models. In a prophylactic study, all the WNZ rabbits having received a dose of 2.5 mg·kg^−1^ of IgG 35PA83 6.20 before an intranasal challenge with 100 LD_50_ of spore survived. In a similar experiment, Raxibacumab was used at a dose of 40 mg·kg^−1^ to reach the same protective results [36]. In a therapeutic study, the same dose of IgG 35PA83 6.20, injected 6 h after a challenge of 80 or 200 LD_50_, allowed a survival rate of 100% and 80%, respectively. These results cannot be compared with those published for Raxibacumab, because therapeutic assays were realized under different conditions. These results suggest that IgG 35PA83 6.20 could be used as a pre-exposure and post-exposure treatment of anthrax, alone or jointly with antibiotics. 35PA83 6.20 may be also an alternative to antibiotics, particularly for the treatment of potential antibiotics resistant strains. IgG 35PA83 6.20 was not tested in another animal model, such as non-human primate, because such studies would have been realized during the pre-clinical development. In conclusion of the in-vivo studies our results suggest that the use of IgG 35PA83 6.20 could be envisioned to improve the pre-exposure and post-exposure treatment of anthrax, when utilized as a complement to antibiotics, or as a solution to potential antibiotics resistant strains.

The in-vitro and in-vivo promising results of Fab 35PA83 6.20 led us to envision its clinical development jointly with the French military structure in charge of the development of military equipment programs, the *Direction Générale de l’Armement* (DGA). Indeed, in France, only the DGA had funding adequate for this clinical development of a medical countermeasure against a disease that is naturally rare and that can be used as a bioweapon. In effect, clinical development includes clinical trials which necessitate, for their phase two, an access to patients to evaluate both tolerance and efficacy of the candidate. It was estimated that access to a sufficient number of patients contaminated by anthrax was an unsurmountable obstacle, due to the low natural incidence (five contaminations in France in 22 years). Health military authorities had to object that biological weapons are, almost by definition, rarely or not encountered naturally, so that standard rules cannot apply for the pharmaceutical development of treatments against biowarfare. It was further insisted that the use of a molecule effective in animal studies (including non-human primates, preferably if pertinent) and well-tolerated in humans was ethically right if vitally needed, as expected for this IgG in the case of inhalational anthrax. This situation in fact corresponds to the phase two clinical trials. DGA thus regarded the development of the IgG 35PA derived from Fab 35PA83 6.20 as legal if limited to the phase one clinical trial, which evaluates both efficacy in animals and tolerance in humans, and financed it. Later, the first use of this IgG would have been regarded as the phase two trial, allowing it to proceed for complete clinical trials. This project, called ATHENA (*Anticorps THErapeutique NEutralisant l’Anthrax*, therapeutic antibody neutralizing anthrax), was the first development of therapeutic antibody by the French ministry of army. Unfortunately, following several industrial and scientific issues, the project was cancelled after 5 years of investment. Here, we reviewed the main difficulties encountered, to provide general information to other academic laboratories that identified a drug candidate and planned a clinical development. Before all the industrial issue, the main difficulty may be to raise the several million euros required to fund a clinical development. The time required to fund a project may be problematic, in part due to the medical field, where the competition is hard. This delay may be problematic, because the first drug marketed have a significant advantage and then it is essential to compare the future drugs with the one marketed. The funding is particularly complex for rare diseases or for the drugs directed against biowarfare agents, such as smallpox, that were eradicated. Generally, there is a scientific gap between the characterization of the antibody realized during the research phase and with the exigences for a clinical development.

To deal with this issue, it is essential to realize study in the Good Manufacturing Practices (GMP) or GMP-like conditions as soon as possible. Early in the project, the intellectual property strategy has to be defined. Particularly, the data essential for the clinical development must be kept confidential, because after any communication such as publication or conference, they will be considered as part of the public domain and thus not patentable. Excepted in some specific cases, such as in USA when the national security is involved, the full sequence of the antibody has to be presented in the patent. When an antibody sequence is published in a patent, it would be possible for competitors to use it for the development of an antibody variant. Industrial secret may represent an alternative to a patent. The management of a clinical project is time-consuming and requires a specific expertise. It may be very difficult for an academic laboratory to realize the management on their own. The identification of a company that is able to do it is essential. Before the beginning of the project, the market targeted by the drug needs to be defined. Indeed, depending on the market, specificity may exist (formulation, vial-filling…). The pre-clinical development includes several mandatory characterizations (stability, PK, efficacy…). Nevertheless, some characterizations should be done before the beginning of the project. For example, in the research phase, glycosylation or fucosylation of antibodies may not be problematic, but such antibody modification, could be problematic when administrated in humans [37]. Identifying and controlling these parameters are essential in modern antibody industry. Indeed, antibody glycosylation is a common post-translational modification that occurs during the production of antibodies. Glycosylation plays an important role in the PK, efficacy, and safety of therapeutics antibodies. Glycosylation sites can be identified in-silico and, if unwanted glycosylation is identified, antibody-engineering can be performed. The low fucose rate of 35PA83 6.20 was a major issue during the project ATHENA. Initially, it was estimated that the antibody binds PA83 in the general circulation. According to this hypothesis, the fucose rate of 35PA83 6.20 would have been a benefit, because the high affinity of 35PA83 6.20 for FcγRIIIAa (CD16a) would have promoted the elimination of the toxin. This hypothesis was questioned during a later epitope mapping study that revealed that the epitope initially identified was not a good one. If the antibody binds PA83 after the binding to the cell surface, it could recruit cytotoxic cells that will lyse the cell. The consequence would be the death of the cell due to an inappropriate immune response instead of the anthrax toxin activity. It would have been possible to engineer the antibody or to express it in a cell line that induces a higher fucose rate. Unfortunately, both options would have delayed the project for several years, because almost all studies would have been completely restarted. The issue that we observed underlines that the precise mechanism of action of the antibody must be precisely defined before the beginning of the preclinical development.

## 4. Materials and Methods

### 4.1. Affinity Determination

The antibody affinities were measured by surface plasmon resonance using a BIAcore 3000 (GE-Healthcare/cytiva), instrument. The toxin was coated at a maximum of 1100 resonance units (RU) on a CM5 chip (GE-Healthcare/cytiva) via amine coupling, according to manufacturer’s instructions. A volume of 100 µL of at least six dilutions of the antibody in HBS-EP buffer (GE-HealthcareMcytiva), were tested. Generally, sample dilution concentrations ranged from 2 µM to 0.1 nM. A 30 µL·min^−1^ flow rate was maintained during the run. After each scFv dilution tested, chip was regenerated with 1.5 µM glycine buffer (GE-Healthcare/cytiva), run for 30 s at 10 µL·min^−1^. Affinities were calculated using the BIAevaluation software (GE-Healthcare/cytiva) according to Langmuir adsorption model and results were verified by internal consistency tests.

### 4.2. Humanization

The human germline antibody-sequence closest to 35PA83 sequence was identified on-line using IMGT^®^ database. The amino acid diverging between 35PA83 and the germline sequence were identified. Each amino acid diverging between the sequences represents a potential hotspot for T-epitope, and should be removed. Antibody variants containing one or several identified mutations were expressed and purified by Proteogenix. The mutations inducing stability issue during the antibody expression were not selected. The affinity of the expressed antibody variants was determined by surface plasmon resonance. The mutations inducing a significant decrease in affinity were not selected. The mutation inducing no or low decrease in affinity were selected and combined in new variants, until a variant containing a maximum of mutation was obtained.

### 4.3. Preparation of Bacillus Anthracis Spores

The spore stocks of the *B. anthracis* clinical isolate 9602 (pXO1+/pXO2+) [38] were from the *Institut Pasteur de Paris* collection. Stock of *B. anthracis* spores were produced and purified on Radioselectan (Renografin 76%, Schering) as previously described [39,40]. They were stored at −20 °C until use and each vial was checked by plating and determination of viable counts after thawing.

### 4.4. ELISA

Ninety-six-well microtiter plates (Nunc Maxisorp, Sigma Aldrich, L’lsle-d’Abeau Chesne, France) were coated by incubation with PA_83_, LF (List laboratories) or KLH (as control; Sigma Aldrich, L’lsle-d’Abeau Chesne, France) diluted in PBS (5 µg/mL, 100 µL per well) overnight at 4 °C. Plates were blocked by incubation for 2 h at 37 °C with 200 µL of 5% BSA in PBS. Sera were serially diluted in 0.1% Tween 20/1% BSA PBS and incubated with the plates (100 µL/well) for 2 h at 37 °C. We then incubated the plates with an anti-mouse IgG alkaline phosphatase conjugate or an anti-human IgG alkaline phosphatase conjugate or an anti-rabbit IgG alkaline phosphatase conjugate (Sigma, St. Louis, MO, USA) (1/10,000) for 1 h at 37 °C. P-nitrophenyl phosphate (Sigma Aldrich) substrate was then added and the plates were incubated for 30 min at room temperature. Absorbance at 405 nm was determined using an automated microplate reader (iEMS reader MF, Labsystems, Helsinki, Finland). The end-point dilution, the reciprocal value of which corresponded to the antibody titer of the serum, was defined as giving a signal equal to twice that of the naive serum, used as a negative control.

### 4.5. Pharmacokinetic Studies

The elimination half-life of IgG 35PA_83_ was evaluated in WNZ rabbits. Six WNZ rabbits (Charles River laboratories, L’Arbresle, France) received 5 mg·kg^−1^ of the antibody by intravenously injection via ear vein. Blood (2 mL) of each animal was collected by sampling in the ear artery at the following time: pre-injection, 0.5 h, 1 h, 3 h, 12 h, 24 h, 72 h, 96 h, 144 h, 216 h, 360 h, 504 h and 672 h. The IgG 35PA_83_ concentrations were determined for all animals and for each bleeding day by ELISA (see above). Pharmacokinetics was evaluated by non-compartmental analysis of the test item plasma concentration data using non validated computer software (WinNonlin, version 5.0, Pharsight Corp., Mountain View, CA, USA). The area under the curve (AUC) was calculated using the linear trapezoidal method (linear interpolation). The terminal elimination phase of the pharmacokinetics (PK) profile was identified and its slope calculated using log-linear regression. The coefficient of determination of the line fitted to the terminal-elimination phase was calculated. PK parameters describing the systemic exposure of the test item in the test system were estimated from observed plasma concentration values, the dosing regimen, the AUC, and the terminal-elimination phase rate constant.

### 4.6. Passive Immunization and Delayed Treatment with IgG 35PA83 Alone in White New Zealand Rabbits Infected by 9602 Spores

For passive immunization study, IgG 35PA83 were injected intravenously (ear vein) at 2.5, 1, 0.5 mg·kg^−1^ in three groups of eight WNZ rabbits, previously anesthetized using imalgene 1000 (Merial, Lyon, France). Five minutes after, animals were challenged by 25 μL of the virulent 9602 *B. anthracis* strain spores 25 suspension deposited on each nare for inhalation into the lungs, and corresponding to 100 LD_50._

For the delayed treatment test, same experimental conditions were used, except that two groups of 8 animals received the IgG injection (2.5 mg·kg^−1^) 6 h after a challenge of 80 LD_50_ or 200 LD_50_ of 9602 *B. anthracis* spores.

For each group, four additional animals were only challenged under the same experimental conditions and utilized as positive control. All experiments using the 9602 strain were performed in a biosafety level 3 containment area, and animals were observed 21 days after challenge.

### 4.7. Statistical Analysis of In Vivo Studies

Log-rank analyses of Kaplan-Meier survival curves were carried out with GraphPad Prism 4.0 software (GraphPad software, San Diego, CA, USA).

## Figures and Tables

**Figure 1 toxins-14-00172-f001:**
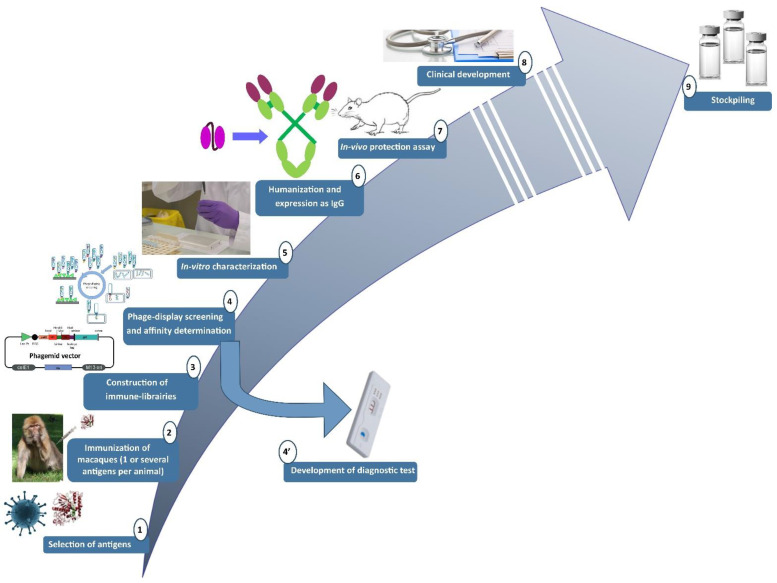
Pipeline used at IRBA for the research and development of recombinant antibodies.

**Figure 2 toxins-14-00172-f002:**
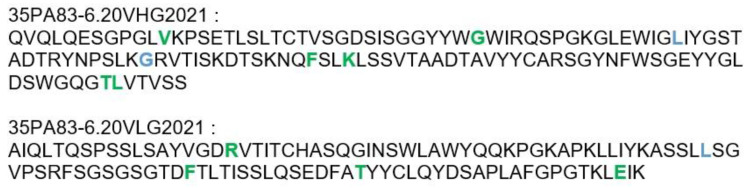
Amino acid sequence of 35PA83 6.20 hyper-humanized. The mutations introduced are colored in green. The mutations resulting from the affinity maturation are colored in blue.

**Figure 3 toxins-14-00172-f003:**
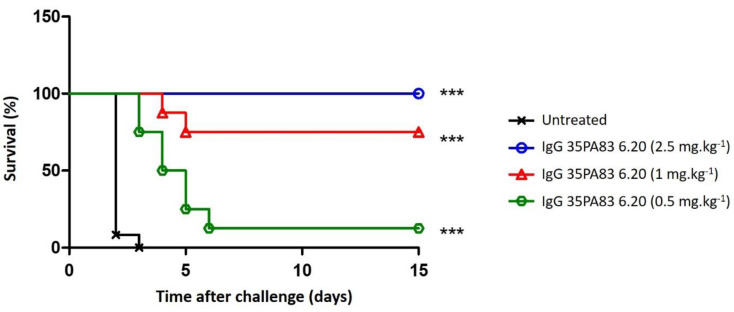
Passive prophylaxis by IgG 35PA83 in WNZ rabbits. One injection of IgG 35PA83 (2.5, 1 et 0.5 mg·kg^−1^) was administered five minutes before the challenge (intranasal) of WNZ rabbits using 100 LD50 of the *B. anthracis* lethal strain 9602. No new event was seen beyond the 15th day. Significant effects are shown with a *** (*p* < 0.0001).

**Figure 4 toxins-14-00172-f004:**
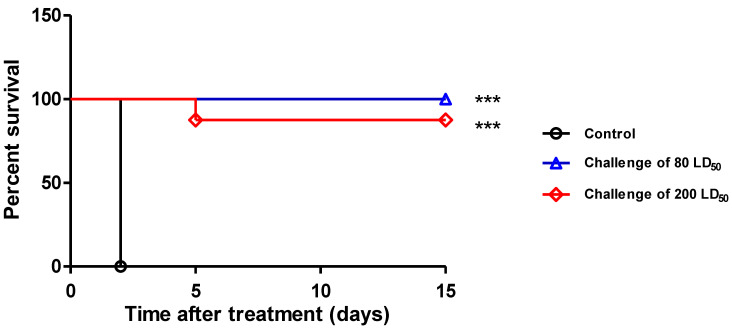
Treatment with IgG 35PA83 alone in WNZ rabbits. One injection of IgG 35PA83 (2.5 mg·kg^−1^) was administered six hours after the challenge (intranasal) of WNZ rabbits using 80 or 200 LD50 of the *B. anthracis* lethal strain 9602. No new event was seen beyond the 15th day. Significant effects are shown with a *** (*p* = 0.0009).

**Figure 5 toxins-14-00172-f005:**
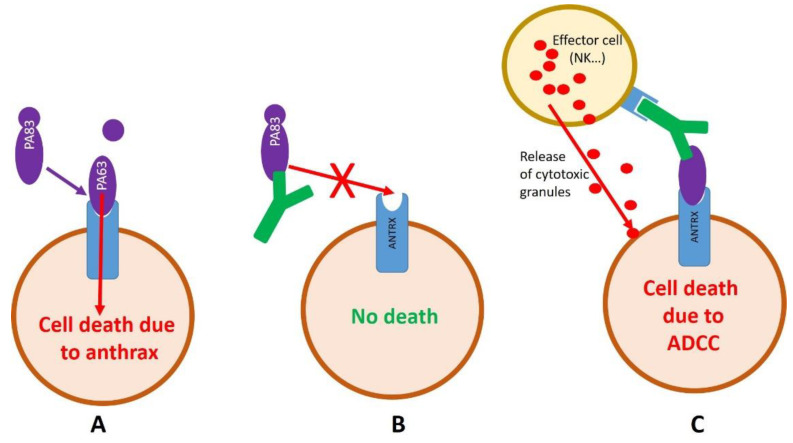
Impact of the epitope localization on cell death. (**A**) In absence of antibody, PA83 binds its cell receptor (ANTRX1/TEM8 or ANTRX2/CMG2) on the cell surface. PA83 is then cleaved and PA20 is released from PA63. After heptamerization and cell-penetration, the cell died due to the toxin activity. (**B**) If the epitope of 35PA83 6.20 is localized in the domain IV of PA83, it can bind it in the global circulation. Such binding will prevent PA83 binding to its receptor and the toxin will be eliminated by the immune system; cell will not die. (**C**) If the epitope is outside domain IV, PA83 can bind the cell receptor and then 35PA83 6.20 can bind PA63 at the cell surface. The antibody can also bind PA83 in the global circulation and then the PA83-antibody complex can bind the cell. Because 35PA83 6.20 is an IgG1, is can bind all FcγR with high affinity and induce an ADCC response. Thus, effector cells, such as NK, may be recruited and kill the cell that bound the PA63-antibody complex.

**Table 1 toxins-14-00172-t001:** Overview of all mutations identified in 2LF and consequences on affinity. The mutation emphasized in red was not selected.

VH			VL		
Mutation	KD (nM)	Region	Mutation	KD (nM)	Region
Q5V	1.72	FR1	V11L	1.99	FR1
L11V	2.09	FR1	R18K	1.64	FR1
A12V	2.09	FR1	R24H	1.99	FR1
K13Q	2.09	FR1	K42N	2.29	FR2
G16R	2.09	FR1	I48L	1.22	FR2
L34M	3.09	FR2	Y49H	1.49	FR2
S49A	0.8	FR2	Q55E	1.32	FR3
K73N	2.17	FR3	S56N	2.6	FR3
K75N	7.54	FR3	T69A	1.85	FR3
V78L	2.96	FR3	F71Y	2.05	FR3
S79V	1.8	FR3	P80S	1.92	FR3
A87S	1.5	FR3	V106I	1.53	FR4
E88D	2.41	FR3			
H94Y	1.09	FR3			
R114Q	2.09	FR4			
V116T	1.57	FR4			
L117M	2.09	FR4			

**Table 2 toxins-14-00172-t002:** Overview of all mutations identified in the complementary determining region of 2LF and con-sequences on affinity. The mutations emphasized in red was not selected, the mutations in green were selected.

VH			VL		
Mutation		Region	Mutation		Region
A29T	1.2 nM	CDR1	K56A	33 nM	CDR2
D35S	1.3 nM	CDR1	Y107A	20 nM	CDR3
T37Y	1.6 nM	CDR1	S108N	0.8 nM	CDR3
T58Y	1.8 nM	CDR2	T109S	0.6 nM	CDR3
G59D	5.5 nM	CDR2	S114F	2.9 nM	CDR3
T64K	0.8 nM	CDR2	I116L	0.8 nM	CDR3
G113D	19 nM	CDR3			
P114A	21 nM	CDR3			
L115F	6 nM	CDR3			

**Table 3 toxins-14-00172-t003:** Overview of all mutations identified in the framework regions of 35PA83 6.20 and consequences on affinity (Koff) and stability. The selected mutations are emphasized in green; the rejected mutations are emphasized in red. Numeration is done according to IMGT^®^ domain gap align standard. “Unstable” referred as antibodies that failed to be expressed with a sufficient titer or that were precipitated.

VH				VL			
Mutation	koff (s^−1^)	Region	Remark	Mutation	koff (s^−1^)	Region	Remark
L13V	1.31 × 10^−4^	FR1		Y14S	4.39 × 10^−4^	FR1	
S40G	1.10 × 10^−4^	FR2		K18R	3.45 × 10^−4^	FR1	
S45P	4.31 × 10^−4^	FR2	Unstable	H24R	4.35 × 10^−4^	FR1	Unstable
K80V	9.08 × 10^−4^	FR3		L68E		FR3	Affinity maturation
L87F	3.73 × 10^−4^	FR3		Y87F	1.29 × 10^−4^	FR3	
Q90K	1.10 × 10^−4^	FR3		S96P	Not determined	FR3	Unstable
R92S	3.01 × 10^−4^	FR3	Unstable	S101T	1.59 × 10^−4^	FR3	
A122T	2.63 × 10^−4^	FR4		D119E	2.80 × 10^−4^	FR4	
V123L	2.93 × 10^−4^	FR4	ND				

**Table 4 toxins-14-00172-t004:** Pharmacokinetic parameters after a single subcutaneous administration of 35PA83 in WNZ rabbit. MRT: mean residence time. VD: Volume of Distribution. Cl: clearance. AUCo-∞: areas under the curves from zero. Cmax: peak concentration. T_1/2_: half-life.

Parameters	WNZ Rabbit
Dose injected (mg·kg^−1^)	5
T _½_ (h)	94.6
MRT (h)	143
AUC_0-∞_ (h·µg·mL^−1^)	3609
C_max_ (µg·mL^−1^)	36.9
Cl (mL·h^−1^·kg^−1^)	1.88
Vd (mL·kg^−1^)	228

## Data Availability

All raw data are the propriety of the French ministry of armies.

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
