# Peer review of "Antibodies against Anthrax Toxins: A Long Way from Benchlab to the Bedside"

_toxins, 2022, doi:10.3390/toxins14030172_

Round 1

Reviewer 1 Report

In the manuscript "Antibodies against anthrax toxins: a long way from bench lab to the bedside," the authors describe their experience with preclinical development of an antibody aimed at neutralising protective antigen of the Bacillus anthracis lethal toxin. Although a bit long, the manuscript is an interesting read, describing several pitfalls one should watch out for when aiming for preclinical and clinical trials. However, several points should be addressed before publication:

1) The 35PA83 6.20 epitope analysis results are problematic in theory, but the experiments with rabbits showed that the antibody works in practice. Thus, can you further argue why the theoretical possibility of inappropriate immune activation precluded further development?
2) In Section 2.1, when describing the pipeline for developing recombinant antibodies, provide more references. True, some references are provided in Section 2.2. Nevertheless, they should also be given the first time a procedure is mentioned. For example, in line 106: "bone marrow was sampled, total RNA was retro-amplified and the DNA region coding the antibody variable domains were amplified with specific primers," please provide detailed protocol or suitable reference for these procedures in and define "RNA retro-amplification" (do authors mean reverse transcription?)
3) lines 97, 198, 206, 239, 283, 296, 304: reference is missing
4) line 107: "no significant amplification" > do authors mean "no significant expression"?
5) line 113: "used for the generation of a phage-display single chain Fragment variable (scFv) or Fragment antigen-binding (Fab) library" > please provide a reference for a similar protocol
6) line 139: please define abbreviations FR and CDR
7) line 229: provide a reference for the Massive Mutagenesis approach
8) line 248: muted > mutated
9) line 284: define abbreviation S.C.
10) line 331: tability > stability
11) line 419-421: the sentence is unclear; please rephrase
12) line 476: sentence fragment
13) line 524: flaconning > packaging
14) mention the PA83 explicitly in the Introduction

Author Response

Dear reviewer,

Please find below our answer to your comments. We are pleased to announce that we were able to answer positively to all your remarks.

In the manuscript "Antibodies against anthrax toxins: a long way from bench lab to the bedside," the authors describe their experience with preclinical development of an antibody aimed at neutralising protective antigen of the Bacillus anthracis lethal toxin. Although a bit long, the manuscript is an interesting read, describing several pitfalls one should watch out for when aiming for preclinical and clinical trials. However, several points should be addressed before publication:

  • The 35PA83 6.20 epitope analysis results are problematic in theory, but the experiments with rabbits showed that the antibody works in practice. Thus, can you further argue why the theoretical possibility of inappropriate immune activation precluded further development?

A figure was added to facilitate the comprehension.

2) In Section 2.1, when describing the pipeline for developing recombinant antibodies, provide more references. True, some references are provided in Section 2.2. Nevertheless, they should also be given the first time a procedure is mentioned. For example, in line 106: "bone marrow was sampled, total RNA was retro-amplified and the DNA region coding the antibody variable domains were amplified with specific primers," please provide detailed protocol or suitable reference for these procedures in and define "RNA retro-amplification" (do authors mean reverse transcription?)

A reference (Arnaud AVRIL, springer protocol 2018) was added. This reference provides a completely described protocol. Retro-amplification was changed to RT-PCR.

3) lines 97, 198, 206, 239, 283, 296, 304: reference is missing

This issue was due to the article formatting by the editor. There is no issue in the submitted manuscript. A special note will be added for the editor.

4) line 107: "no significant amplification" > do authors mean "no significant expression"?

To clarify the sentence was modified:  no significant amplification of the gene coding for the antibody variable genes is expected in RT-PCR

5) line 113: "used for the generation of a phage-display single chain Fragment variable (scFv) or Fragment antigen-binding (Fab) library" > please provide a reference for a similar protocol

References were added.

6) line 139: please define abbreviations FR and CDR

Done.
7) line 229: provide a reference for the Massive Mutagenesis approach

Done.

8) line 248: muted > mutated

Done.

9) line 284: define abbreviation S.C.

Done.
10) line 331: tability > stability

Done.

11) line 419-421: the sentence is unclear; please rephrase

Done.

12) line 476: sentence fragment

The issue was not found in the original text ; it was due to journal editing. 

13) line 524: flaconning > packaging

Done.
14) mention the PA83 explicitly in the Introduction

Done.

Reviewer 2 Report

Very interesting and well written.  

Author Response

Dear reviewer,

We are pleased to announce that we were able to answer positively to all your remarks.

Reviewer 3 Report

The manuscript entitled "Antibodies against anthrax toxins: a long way from benchlab to the bedside" left a mixed impression on the reviewer. On the one hand, the authors describe their way of developing anti-anthrax therapeutic antibodies from the formulation of the problem to preclinical development. This may be of interest to the reader. On the other hand, the text pays too little attention to the scientific side of the issue, and it remains unclear how the results obtained by the authors differ from the results obtained by other researchers. This reduces the scientific value of the manuscript. In addition, in the text of the manuscript, the authors repeatedly say that they received certain results, but note “data not shown”. It seems to me that this is not quite correct, it would be better to give a link to the relevant publications, or add the results to this manuscript as supplementary materials.

In addition to mentioning the main shortcomings, I would like to make a few minor remarks.

  1. Lines 23-24

Anthrax is a human lethal disease caused by the spore-forming, Gram-positive bacterium Bacillus anthracis

The statement is not well formulated. Anthrax can affect humans, but it is not a specific human disease. In nature, anthrax affects mainly ungulates.

  1. Lines 25-26

Each of these are lethal for humans in absence of treatment

This statement is worth rephrasing. Cutaneous anthrax is relatively rarely fatal. The authors themselves indicate this a little lower in the text.

  1. Lines 47-48

In the past several countries such as Russian Soviet Federative Socialist Republic, the Great Britain, Canada, the United States of America and Iraq were suspected to weaponized anthrax.

This statement is not correctly formulated. Russian Soviet Federative Socialist Republic was not the only republic included in the USSR. Therefore, it is not entirely correct to attribute to the RSFSR any action taken by the USSR, it looks like an attempt to politicize the issue, which is inappropriate in a scientific publication

  1. Lines 56-57

Additional limitations of antibiotics include poor patient compliance with the 60-day schedule and inactivity against antibiotic-resistant strains of B. anthracis.

This assertion deserves to be supplemented somewhat. The reviewer is not aware of natural strains of B. anthracis that are resistant to antibiotics used for the treatment of anthrax. Most likely, we are talking here only about weaponized strains with artificial antibiotic resistance

  1. Lines 96-97

Editorial note left in the text «(Error! Reference source not found.)»

  1. Line 146

Fig. 1 is not of high resolution enough to make some of the text displayed on it unreadable. I hope that the resolution will be higher in the full version of the article.

  1. Lines 170-172

As anthrax is a biowarfare agent and was militarized in the past, it can be envissioned that PA would be intentionally mutated to make inefficient the antibodies currently developed

This statement looks very speculative. The variability in the sequence of PA among natural strains of B. anthracis does not appear to be sufficient for antibody therapy to be ineffective for any strain. The suggestion that the PA sequence will be artificially altered so that the protein remains functionally active but not recognized by antibodies seems very doubtful. At least the reviewer is not aware of such cases. The reason given by the authors for why the PA sequence could be artificially altered also seems dubious. The goal of creating a strain that causes a disease that is difficult to treat can be achieved much more simply, for example, by introducing antibiotic resistance genes into the genome. This is not only easier, but also more logical, since antibiotics are used much more widely than specific antibodies, and during an outbreak of the disease they will be delivered to the affected contingent much earlier.

  1. Lines 170-172

The same as in Remark 5. The same applies to Lines 199, 206,239-240 283-284, 297, 304

  1. Line 205

The abbreviation CDR is mentioned in the text for the first time without decoding. The decryption is given a few lines below.

  1. Lines 288-291

Table 4 is not very clearly presented. It is not entirely clear why the table header contains “Dose injected” if the value of this dose is not indicated

  1. Line 331

The letter «s» is missing from the word «stability»

  1. Lines 333-334

The exponent was moved to another line. Should be reformatted

  1. Line 336

It is not entirely clear why Isoelectric focusing is written in italics in the list of methods used

  1. Line 476

The sentence is not finished

Author Response

Dear reviewer,

Please find below our answer to your comments. We are pleased to announce that we were able to answer positively to all your remarks.

The manuscript entitled "Antibodies against anthrax toxins: a long way from benchlab to the bedside" left a mixed impression on the reviewer. On the one hand, the authors describe their way of developing anti-anthrax therapeutic antibodies from the formulation of the problem to preclinical development. This may be of interest to the reader. On the other hand, the text pays too little attention to the scientific side of the issue, and it remains unclear how the results obtained by the authors differ from the results obtained by other researchers. This reduces the scientific value of the manuscript. In addition, in the text of the manuscript, the authors repeatedly say that they received certain results, but note “data not shown”. It seems to me that this is not quite correct, it would be better to give a link to the relevant publications, or add the results to this manuscript as supplementary materials.

In addition to mentioning the main shortcomings, I would like to make a few minor remarks.

  1. Lines 23-24

Anthrax is a human lethal disease caused by the spore-forming, Gram-positive bacterium Bacillus anthracis

The statement is not well formulated. Anthrax can affect humans, but it is not a specific human disease. In nature, anthrax affects mainly ungulates.

Correct. The sentence was clarified.

  1. Lines 25-26

Each of these are lethal for humans in absence of treatment

This statement is worth rephrasing. Cutaneous anthrax is relatively rarely fatal. The authors themselves indicate this a little lower in the text.

Correct, the lethality of cutaneous anthrax is much lower lethal than pulmonary anthrax. Nevertheless, all forms of anthrax are lethal. Text was slightly modified.

  1. Lines 47-48

In the past several countries such as Russian Soviet Federative Socialist Republic, the Great Britain, Canada, the United States of America and Iraq were suspected to weaponized anthrax.

This statement is not correctly formulated. Russian Soviet Federative Socialist Republic was not the only republic included in the USSR. Therefore, it is not entirely correct to attribute to the RSFSR any action taken by the USSR, it looks like an attempt to politicize the issue, which is inappropriate in a scientific publication.

Correct. The intention is not to name and shame some states. The objective is only to give example of previous programs. The examples given are only the ones that were officially proven by the international community. The potential confusion between RSFSR and USSR was clarified in the text.

  1. Lines 56-57

Additional limitations of antibiotics include poor patient compliance with the 60-day schedule and inactivity against antibiotic-resistant strains of B. anthracis.

This assertion deserves to be supplemented somewhat. The reviewer is not aware of natural strains of B. anthracis that are resistant to antibiotics used for the treatment of anthrax. Most likely, we are talking here only about weaponized strains with artificial antibiotic resistance.

Correct. Clarified in the text.

  1. Lines 96-97

Editorial note left in the text «(Error! Reference source not found.)»

This issue appeared after text editing by the editor. A special note to the editor was added.

  1. Line 146

Fig. 1 is not of high resolution enough to make some of the text displayed on it unreadable. I hope that the resolution will be higher in the full version of the article.

We hope so, because the quality of the original figure is sufficient. A special note was added for the editor.

  1. Lines 170-172

As anthrax is a biowarfare agent and was militarized in the past, it can be envissioned that PA would be intentionally mutated to make inefficient the antibodies currently developed

This statement looks very speculative. The variability in the sequence of PA among natural strains of B. anthracis does not appear to be sufficient for antibody therapy to be ineffective for any strain. The suggestion that the PA sequence will be artificially altered so that the protein remains functionally active but not recognized by antibodies seems very doubtful. At least the reviewer is not aware of such cases. The reason given by the authors for why the PA sequence could be artificially altered also seems dubious. The goal of creating a strain that causes a disease that is difficult to treat can be achieved much more simply, for example, by introducing antibiotic resistance genes into the genome. This is not only easier, but also more logical, since antibiotics are used much more widely than specific antibodies, and during an outbreak of the disease they will be delivered to the affected contingent much earlier.

Of course, the PA is naturally stable and the risk of intentional mutation is limited. Intentional mutation of PA will be technically complicated, but with a State budget, such limitation will not be an issue.  Nevertheless, the role of biodefense is to be prepared for all scenario, even if it appeared unlikely. An additional sentence was added in the text.

  1. Lines 170-172

The same as in Remark 5. The same applies to Lines 199, 206,239-240 283-284, 297, 304

Done.

  1. Line 205

The abbreviation CDR is mentioned in the text for the first time without decoding. The decryption is given a few lines below.

Done.

  1. Lines 288-291

Table 4 is not very clearly presented. It is not entirely clear why the table header contains “Dose injected” if the value of this dose is not indicated.

This is correct. The editing issue was fixed.

  1. Line 331

The letter «s» is missing from the word «stability»

Done.

  1. Lines 333-334

The exponent was moved to another line. Should be reformatted

No exponent observed in these lines.

  1. Line 336

It is not entirely clear why Isoelectric focusing is written in italics in the list of methods used.

Correct. Text formatted.

  1. Line 476

The sentence is not finished

Issue not observed. It should be an issue after journal editing. A special note was added for the editor.
